# The Role of Distance from Home to Hospital on Parental Experience in the NICU: A Qualitative Study

**DOI:** 10.3390/children10091576

**Published:** 2023-09-20

**Authors:** Stephanie L. Bourque, Venice N. Williams, Jessica Scott, Sunah S. Hwang

**Affiliations:** 1Department of Pediatrics, Section of Neonatology, University of Colorado School of Medicine, Aurora, CO 80045, USA; jessica.scott@cuanschutz.edu (J.S.); sunah.hwang@childrenscolorado.org (S.S.H.); 2Department of Pediatrics, Prevention Research Center for Family & Child Health, University of Colorado School of Medicine, Aurora, CO 80045, USA; venice.williams@cuanschutz.edu

**Keywords:** family engagement, prematurity, geographic factors, neonatal intensive care unit

## Abstract

Prolonged admission to the neonatal intensive care unit presents challenges for families, especially those displaced far from home. Understanding specific barriers to parental engagement in the NICU is key to addressing these challenges with hospital-based interventions. The objective of this qualitative study was to explore the impact of distance from home to hospital on the engagement of parents of very preterm infants (VPT) in the neonatal intensive care unit (NICU). We used a grounded theory approach and conducted 13 qualitative interviews with parents of VPT who were admitted ≥14 days and resided ≥50 miles away using a semi-structured interview guide informed by the socio-ecological framework. We used constant comparative method with double coders for theme emergence. Our results highlight a multitude of facilitators and barriers to engagement. Facilitators included: (1) individual—delivery preparedness and social support; (2) environmental—medical team relationships; and (3) societal—access to perinatal care. Barriers included: (1) individual—transfer stressors, medical needs, mental health, and dependents; (2) environmental—NICU space, communication, and lack of technology; and (3) societal—lack of paid leave. NICU parents with geographic separation from home experienced a multitude of barriers to engagement, many of which could be addressed by hospital-based interventions.

## 1. Introduction

More than 10% of U.S.-born infants are born prematurely and experience a birth hospitalization of weeks to months [1,2]. Families experience limitations on frequency and quantity of care they can provide to their infants during neonatal intensive care unit (NICU) admission. Individual and systemic barriers to engagement include the stress and trauma of preterm delivery, the physical environment of the NICU, socioeconomic stressors, limited English proficiency, and lack of paid leave [3,4,5]. These factors limit the ability to which families can engage in family-centered care (FCC), an approach to medical care based on the belief that optimal health outcomes are achieved when family members are active participants in providing social, emotional, and developmental support [6]. In the NICU, FCC supports and allows parents to participate more fully in caring for and making medical decisions for their infants.

Qualitative research has identified barriers to FCC including caregiving responsibilities and lack of family support [3,4]. The positive impacts of integrating parents is well established, as in, for example, through processes such as kangaroo care and family integrated care [7,8]. However, among families who are geographically displaced from home during their infant’s NICU admission, little is known specifically about the impact of distance on the factors that affect a parent’s ability to be physically present to provide care to their infant in the NICU [7,8,9].

Previous work has demonstrated an inverse association between both distance and travel time to the hospital with parental presence in the NICU [10]. Factors including postpartum medical issues, financial strain, and employment obligations may be more pronounced with geographic separation [11,12,13]. For families residing further away and in rural locations, preterm delivery and postnatal transfer are common and opportunities for back-transport closer to home are limited [14,15]. We hypothesized that families who are geographically displaced from home will disproportionally experience barriers and challenges that are different compared to the general NICU population.

Our study objective was to explore facilitators and barriers that families of very preterm infants (≤32 weeks gestational age, VPT) encounter with physical presence and engagement in the NICU, when geographically displaced from their primary residence.

## 2. Methods

### 2.1. Study Design

Using a socio-ecological conceptual framework, this qualitative study examined the impact of distance from home on the parental experience during a prolonged birth hospitalization [16]. We adapted grounded theory by using this framework, illustrated in Figure 1, to inform development of the interview guide and overall reporting of results. Rather than implementing theoretical sensitivity as typical in traditional grounded theory, we entered the research setting with predetermined ideas of parental engagement in the NICU setting. In this study, we explored: (a) individual-level factors associated directly with the parent, infant or parent-infant dyad; (b) environmental-level factors including the physical NICU space, interaction with medical providers and unit-level policies impacting parental engagement in the NICU; and (c) societal-level factors including the impact of state policies and availability of community support programs.

### 2.2. Participants, Context, and Recruitment

Inclusion criteria included biologic or adoptive parents of VPT infants born at ≤32 weeks gestation and whose primary residence was ≥50 miles from the hospital. Participants were recruited from a regional level IV NICU, situated on an academic medical campus with a multi-state catchment area. Location of birth was defined as inborn, preterm infants born in the hospital with the level IV NICU or outborn, preterm infants born elsewhere and transferred postnatally. Urban or rural residence was defined by county at time of delivery. Urban counties are defined by the Office of Management and Budget (OMB) as those with an urban core of 50,000 or more people and rural counties are defined as those that are not urban.

Recruitment was conducted consecutively and families who had been admitted for at least 14 days were approached for consent via telephone. Fourteen days was chosen to provide sufficient NICU exposure and time beyond the acute admission experience for families. A total of 18 participants met eligibility criteria, of whom 72% consented to participate. Participants were compensated for participation with a USD 40 gift card and this study was approved by the Colorado Multiple Institutional Review Board.

### 2.3. Data Collection

A semi-structured interview guide used the socio-ecological model to evaluate the individual, environmental, and societal factors that impacted parents’ NICU experience [17,18]. The interview guide was piloted by parent research partners and edited to ensure it included relevant topics (Table 1). The interviews were conducted by a qualitatively trained Ph.D. research team member who did not provide medical care to the participants’ infant(s). In-depth interviews were conducted remotely by phone or video. Interviews were conducted until saturation was achieved, where no new information was shared by participants. Interviews were transcribed for analysis using an independent transcription service.

### 2.4. Analysis

Transcripts were validated and inputted to NVivo 12 [19]. We used open, axial, and selective coding to allow themes to initially emerge, inform adjustments to the socio-ecological framework as it relates to this study, and develop a better understanding of facilitators and barriers for families of VPT infants to engage in the NICU when displaced from their primary residence [20]. Two coders (the interviewer and a neonatologist health services researcher) coded inductively to develop an iterative codebook. We participated in coding consistency meetings, utilizing the kappa statistic (with a benchmark greater than 0.6) to assess inter-rater reliability and to guide discussions to reach consensus on code definitions [21]. We double coded three transcripts (23%) to assess consistency and then distributed remaining transcripts for individual coding. The primary coder wrote a memo for each transcript to ensure retention of ideas and to synthesize concepts. The research team then generated memos for main themes across participants. Several steps were taken to ensure rigor in the analysis. The research team, including clinical and methodological experts, used an iterative approach to analysis to adjust interview questions including more targeted questions about the impact of distance, develop themes, and monitor for thematic saturation. We triangulated information in our data collection and conducted peer debriefing with NICU staff to elicit feedback on our findings and to allow for researchers to be reflexive in the research process, questioning our own potential biases and assumptions [22].

## 3. Results

### 3.1. Participant Characteristics

Demographics of the thirteen participants are outlined in Table 2. Overall, participants’ primary residence was an average of 285 miles from the hospital, ranging from 51 to 1643 miles distance. Most participants lived in state (77%), while 23% participants lived out of state.

Below, we present facilitators followed by barriers to engagement at the individual, environmental, and societal levels with illustrative quotes outlined in Table 3 and Table 4. Individual facilitators related to preparedness for delivery and readiness for discharge; environmental facilitators included relationships with the medical team and use of technology, while societal factors were related to post-discharge services. Postnatal transfers, infant medical care, perinatal mental health, and readiness for discharge were individual-level barriers, while environmental barriers included the physical NICU environment, other dependents, the medical team’s inconsistency, and lack of access to technology; societal barriers were paid leave and unfamiliarity with post-discharge services.

#### 3.1.1. Theme 1: Individual Facilitators to Engagement

Preparedness for Delivery: Compared to parents of inborn infants, parents of outborn infants had an overall more heterogeneous prenatal experience including prenatal counseling and antepartum admission. For inborn infants, the antepartum period allowed for introduction to the NICU and enabled an increased sense of preparedness for delivery.

Social Support: Almost all participants discussed their social supports, especially among those who lived far away from the hospital. Participants highlighted how these individuals provided additional tangible support with food, financial resources, and childcare, and served as a primary source of emotional support. For those who lived far away, social supports available through the hospital, like the chaplain and other NICU parents, were particularly helpful.

Readiness for Discharge: Nearly all participants discussed how they were preparing for discharge from an emotional and technical standpoint. Many discussed how their time in the NICU had prepared them for home by better understanding the needs of their infant, how to care for them and what to anticipate post discharge. The impact of distance was especially notable for families who resided out of state or several hundred miles away and were less likely to receive consistent follow-up care at the regional children’s hospital.

#### 3.1.2. Theme 2: Environmental Facilitators to Engagement

Medical Team/Ancillary Staff: Beyond individual facilitators to engagement, many participants discussed the guidance and support they had received from their medical team at the hospital. Some explained how hospital staff, in particular bedside nurses, had supported them in becoming comfortable with their infant’s care. This ranged from education from lactation to nurses understanding a parent’s comfort level with conducting infant cares, feeding, and skin-to-skin.

Technology: The use of technology for communication was mentioned by many participants and included the use of video chat and the power of social media to engage with additional community supports. A few participants who lived far away shared that they appreciated the ability to check the MyChart application while not at the bedside for updated medical information, as they were not always able to be physically present in the NICU.

#### 3.1.3. Theme 3: Societal Facilitators to Engagement

Post-discharge Services: All participants shared about their infant’s medical needs when transitioning to care at home and anticipating the need for specialized equipment and services, such as therapists and pediatric subspecialists, regardless of geography. Out-of-state participants also shared perspectives regarding the importance of available therapy services to support their infant’s development.

In addition to facilitators, we heard about many barriers to parental physical presence and ability to engage in the NICU.

#### 3.1.4. Theme 4: Individual Barriers to Engagement

Postnatal Transfer: Participants who delivered at an outside hospital and experienced subsequent transfer had more heterogenous experiences than those with inborn infants. Experience ranged from emergent and immediate postnatal transfer to a planned transfer for subspeciality consultation. One participant described the sudden postnatal transfer to be worrisome and difficult because they were uncertain if their child survived the journey.

Infant Medical Care: Participants who were unable to stay full-time at the hospital and lived further away from the hospital described emotional difficulty in feeling comfortable with their infant’s care while away from the bedside. This difficulty often stemmed from not being able to quickly be present with their infant.

Perinatal Mood and Anxiety Disorders (PMADs): Many participants discussed feelings of depression, anxiety, grief, guilt, and isolation while in the NICU. Several participants noted that it was mentally challenging to engage and bond with their infant, and thus were unable to realize their parental role and lacked control in caring for their infant.

Readiness for Discharge: While readiness for discharge was identified as a facilitator for engagement by some participants, others noted feeling nervous as they prepared for the transition home. These feelings related to not having their in-person nursing support team or the aid of cardiopulmonary monitors, especially if they lived further away from a clinic or hospital.

#### 3.1.5. Theme 5: Environmental Barriers to Engagement

Physical NICU Environment: Many participants shared that while single patient rooms allowed for privacy, the physical setup was not conductive for a prolonged stay both in terms of comfort and noise level. An overall lack of understanding and information regarding the use of machines and wires attached to the infants was cited as a source of stress.

Presence of Other Dependents: For participants with other children, restrictions on sibling visitation during the COVID-19 pandemic impacted the time they were able to spend with their infant, resulting in shorter or infrequent visits. This was shared as a challenge both emotionally and with regards to being able to be physically present for all their children.

Medical Team: While the support of the medical team was described as a facilitator for engagement for some participants, others noted how the medical team could also present barriers for their engagement. Several participants commented on the frustration with having an inconsistent nursing and physician team caring for their infants, often resulting in a lack of continuity of care. For example, one participant commented that with an unfamiliar nurse, she had missed dressing her infant in clothes for the first time, an experience she had been anticipating for weeks.

Access to Technology: A commonly discussed suggestion among participants was optimizing the use of technology for times when parents are not able to be physically present in the NICU. Several participants would have appreciated the opportunity to visit their infant virtually through secure webcam or video.

#### 3.1.6. Theme 6: Societal Barriers to Engagement

Employment and Paid Parental Leave: All participants shared about societal barriers to engagement and regardless of their geography, discussed their or their partner’s current employment situation. Among those who were working prior to delivery, most participants expressed accessing unpaid time off or applying for short-term disability for either themselves or their partners.

Post-Discharge Services: While all participants shared about their infant’s medical needs when transitioning to care at home, some identified post-discharge services as potential barriers to engagement. These included insurance challenges and availability of durable medical equipment and services such as outpatient therapy. One participant noted that hospital staff were not as familiar with out-of-state referrals and establishing necessary insurance coverage for these patients.

## 4. Discussion

This study aimed to explore facilitators and barriers that families of VPT infants in the NICU encounter with physical presence and engagement, when geographically displaced from their primary residence. We describe several facilitators and barriers to family engagement when displaced far from home during a prolonged birth hospitalization, which, for this population, appear to be magnified compared to the general NICU population. Within the socio-ecological framework, we found preparedness for delivery, social support, and readiness for discharge to be facilitators to engagement at the individual level. Environmental facilitators to engagement were related to relationships with the medical team/ancillary staff and use of technology. Societal facilitators were related to post-discharge services and access to outpatient subspeciality care. Conversely, individual barriers included fears and anxiety around postnatal transfer. Environmental barriers were related to challenges of the physical NICU environment, other dependents, and lack of access to technology. Societal barriers included employment, paid parental leave challenges, and availability and access to post-discharge services.

At the individual level, communicating expectations surrounding preterm delivery and potential transfer, while often challenging to predict, can help parents prepare for what is to come. Providing timely information with updates as clinical circumstances change and engaging in honest communication are qualities regarded as important for parents [23]. Social support for families experiencing geographic separation from home is paramount as many are disconnected from their usual social networks and are faced with unprecedented medical, family, and logistical challenges. Specifically, screening and addressing PMADs is crucial, especially for families who are displaced from home and less likely to access primary care or obstetric providers known to them. At baseline, PMAD prevalence in the preterm population exceeds that of the term population and is often under- and undiagnosed [24,25]. Hospital-based initiatives including universal screening during admission and integration of mental health professionals are opportunities to identify and provide resources to this high-risk population who may specifically benefit from telehealth or local referrals close to the NICU [26,27,28].

At the environmental level, the NICU layout plays a key role in parental ability to be physically present and engaged. Numerous studies have demonstrated the benefits of single-family rooms versus open-bay NICUs including increased breastfeeding rates and skin-to-skin time and decreased rates of sepsis [29,30]. While single-family rooms are associated with decreased parental stress, they may contribute to parental isolation and a decreased sense of community within the unit [31]. Maintaining the benefits of single-family rooms and encouraging peer support is one approach to supporting families, especially for those experiencing geographic separation from their home network. This concept was supported by parents in our study, many of whom used their infant’s NICU room as a home away from home if they were unable to frequently travel back to their primary residence.

As technology gains sophistication and accessibility, there is growing pressure to integrate this technology at the bedside [32]. The integration of live video for families to view their infant while not physically present in the NICU, is not currently available in our unit. The use of video visits has been associated with decreased parental stress during periods of infant separation and increased perceived parental involvement [33]. Furthermore, the use of video visits throughout NICU admission resulted in increased provision of breastmilk at discharge [34]. Using video conferencing technology to include parents not physically present in daily medical rounds may be one strategy to overcome geographic barriers for families and to include all caregivers in decision making. The use of technology to support families during admission to the NICU has been well studied internationally with use of secure video messaging systems and live video streaming [35,36,37].

From a societal perspective, achieving equitable paid leave for parents remains an urgent national priority. Benefits of paid leave include decreased odds of both infant and maternal readmission along with improved postpartum stress management [38]. Beyond paid leave, employment policies including flexible hours and remote working are opportunities to allow families of hospitalized infants to be physically present and engage with their infant while remaining in the workforce. This is particularity salient for families who live further away from the hospital, making commuting on a regular basis impracticable. We acknowledge that this impact differs based on local and national paid leave policies and impacts families differently depending on their residence. For example, families residing in countries with well-supported family leave policies may not experience these same barriers to engagement during NICU admission.

Attention to social determinants of health is imperative, especially for families who reside further away from the hospital and are more likely to experience higher expenses and challenges associated with spending time with their infant in the NICU [39]. Childcare, food and housing instability, transportation challenges and employment are infrequently assessed, leaving a gap for resource referral and follow-up. Inclusion of universal and comprehensive screening for SDoH during a prolonged NICU admission is critical to systematically identifying at-risk families [40].

## 5. Limitations

This study is limited by consecutive sampling and is not intended to reflect the experiences of all families of VPT infants. Results may vary for infants cared for in a different NICU setting with different access to resources, degree of family-centered care or social supports. As an exploratory study, it may contain inherent biases including sampling bias towards parents who felt strongly about sharing their experience. Despite these limitations, our study offers nuanced insight into the experiences of parental engagement, particularly for parents who are geographically displaced from their homes, grounded in lived experiences and supported by rigor in our conduct of double coding, data triangulation, and expert debriefs.

## 6. Conclusions

Participants identified several areas of improvement that have potential to specifically impact the experience for families who reside further away from the hospital. These include access to technology and improved communication around discharge. With the identification of these gaps, there is an opportunity for hospital-based interventions to improve parental engagement for this population.

## Figures and Tables

**Figure 1 children-10-01576-f001:**
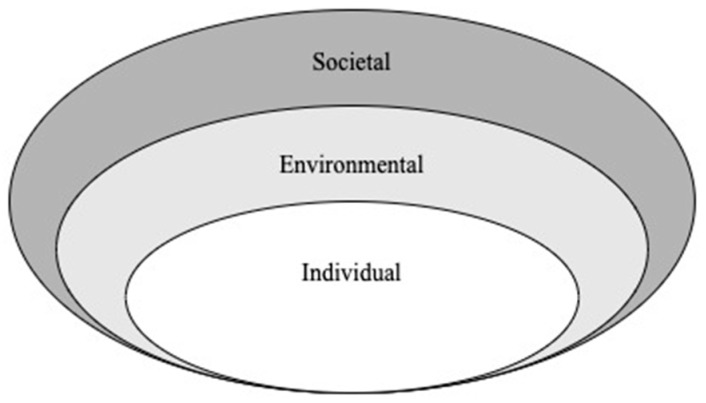
Socio-ecological conceptual model.

**Table 1 children-10-01576-t001:** Abbreviated semi-structured interview guide.

Topic	Sample Question(s)
Hospital Experience	Why was your baby admitted/transferred to the NICU? If transferred, how was the transfer experience for your family?
Challenges Related to Distance	Where do you usually spend the night? How has the distance between hospital and home impacted this decision?
	How has communication between staff been when you have not been physically present in the NICU?
Family Structure	Who is your strongest support person? How is this person helpful?
Financial Implications	What specific costs have you incurred being far from home? Have you had any unanticipated expenses?
Discharge and Follow-Up Care	What challenges do you anticipate facing following hospital discharge?

**Table 2 children-10-01576-t002:** Participant demographics and infant characteristics.

Characteristic	*n* (%)
Birth Location	
Inborn	7 (54%)
Outborn	6 (46%)
Primary Residence Distance (miles) ^†^	285 (51–1643)
Primary Residence County	
Urban	10 (77%)
Rural/Frontier	3 (23%)
Other Dependents Yes No	6 (46%) 7 (54%)
Infant Gestational Age (weeks) ^†^	27 (24–31)
Infant Birthweight (grams) ^†^	1207 (700–1740)

^†^ mean (range).

**Table 3 children-10-01576-t003:** Illustrative quotes: facilitators.

Theme: Subtheme	Quote	Participant Characteristics
Individual: Preparedness for Delivery	*“At our very first appointment, before we even delivered, we were assigned a social worker. That’s been nice, just to know who we would be working with from the get-go before baby was here.”*	#6: inborn, rural, 271 miles
Individual: Social Support	*“He’s just been like, “Hey, are you feeling sad? I understand. Just remember, she’s fine. Do you want me to call and see how she’s doing?” Sometimes I wanna call, but I can’t talk because I’m ugly crying and you can’t understand me. So he’ll call and check how she’s doing and put it on speaker. Then I feel better afterwards. He’s just been very supportive.”*	#10: inborn, urban, 71 miles
Individual: Readiness for Discharge	*“We’re trying to just focus on him and prepare ourselves for life at home. I’m feeling more prepared right now but if you would have asked me a few weeks ago, I’d still be scared of the thought of going home without a nurse. Especially going so far from here. It’s not just right down the road or in the same city. We’re going two states away.”*	#1: outborn, urban, 561 miles
Environmental: Medical Team/Ancillary Staff	*“The nurses are so helpful, they’ll let you decide if you want to pick him up or if you want them to hand him to you to hold. We do skin on skin every day when I’m there, and my boyfriend and I learned how to be pro diaper changers.”*	#8: inborn, urban, 271 miles
Environmental: Technology	*“I love the apps, the iPad bedside app and MyChart. For someone who likes to be informed and wants to use that information to know what’s going on at all times. Especially when we’re gone, I can check his measurements at night, and then I know that information to ask the right questions. Getting his test results immediately is really empowering and encouraging to be on the same page as the doctors and the nurses.”*	#3, outborn, rural, 285 miles
Societal: Post-discharge Services	*“One of my biggest worries and one of my future goals was to make sure that he was getting physical and occupational therapy. Knowing that he is at risk for not reaching milestones. The fact that we already have that here and it’s going to be carried over when we get back to Montana. It might look a little bit different but it is very helpful for us to feel prepared for when we go home.”*	#1: outborn, urban, 561 miles

#: participant identification number.

**Table 4 children-10-01576-t004:** Illustrative quotes: barriers.

Theme: Subtheme	Quote	Participant Characteristics
Individual: Postnatal Transfer	*Honestly, it was awful. It was one of those nerve-wracking moments where you just don’t know. You don’t know if he’s still alive. It was one of the scariest moments for me, just because I didn’t get to see him but for maybe five seconds before he left.”*	#3: outborn, rural, 285 miles
Individual: Infant Medical Care	*“Just not being there for some of his moments. He’s about to move into a crib and we’ll probably miss that, and then he’s had a couple of episodes of the breathing apnea, where he holds his breath and needs stimulation to remind him to breathe. That’s scary, just not being there at all times and not being close by. I can’t just hop in the car and go see him, it’ll take a while to get there.”*	#8: inborn, urban, 71 miles
Individual: Perinatal Mood and Anxiety Disorders	*“Obviously, I’m attached to my baby, but only seeing him for a couple hours a day and not really having control over him, and not feeling like the nurse…It just almost feels like when I go there, I love him, but it’s almost like it’s not my baby yet because I’m not the one looking after him.”* *“I feel emotionally, I have struggled with depression, and I did talk to my doctor about it, and he told me everything that I’m feeling is normal. He’s like, ‘I don’t think it’s related to postpartum depression. I think it’s just the situation you’re in is hard.’ I was prescribed some anti-depressants because it’s not gonna be a two-week thing.”*	#11: inborn, urban, 59 miles #3: outborn, rural, 285 miles
Individual: Readiness for Discharge	*“I’m nervous as hell. I’m a little scared, I ain’t gonna lie. I don’t know what to expect. I hope I can just call the hospital if I have questions, and they can walk me through the steps because it is gonna be completely different being home and on my own.”* *“Definitely not having help right there. Because in the NICU I feel safe because the nurses are right there to jump in if anything happened. We’ll live 20 min away from his doctor, so that’s a little scary. Just knowing that we’re on our own.”*	#4, outborn, rural, 137 miles #8: inborn, urban, 71 miles
Environmental: Physical NICU Environment	*“I would have loved more information about the NICU and what each machine is for so I would understand the beeping. All the machines would beep and I wouldn’t know what it was for. It’s hard because you think something’s happening to your baby.”*	#12: inborn, urban, 70 miles
Environmental: Presence of Other Dependents	*“Siblings can’t come in the NICU. I wish that wasn’t a thing and I could have my daughter here with me and then I could be able to see all my kids in the same day.”*	#6: inborn, rural, 271 miles
Environmental: Medical Team	*“Having a new nurse every day can be exhausting because you have to learn their routine and personality. At least for us, we wish we could have just a few days in a row of familiar faces that we had already built relationships with and with our baby.”*	#7: inborn, urban, 65 miles
Environmental: Access to Technology	*“Having a camera in the room and knowing that there’s certain times in the day that you can just connect to the video camera and see them.”*	#12: inborn, urban, 70 miles
Societal: Employment and Paid Parental Leave	*“My job currently doesn’t offer maternity leave. I did apply for short-term disability, which only covers six weeks unfortunately, depending on how my daughter does from the time she was born until her due date, it’s a total of 14 weeks.”*	#2: outborn, urban, 51 miles
Societal: Post-Discharge Services	*“Being on Wyoming Medicaid, I’m not sure if that’s gonna be as smooth as it would be if I was an in-state resident. I’m assuming it’s not gonna be super-smooth. It’s something that I’m worried about as far different state regulations and different limitations.”*	#4: outborn, rural, 137 miles

#: participant identification number.

## Data Availability

Transcript data from the qualitative interviews conducted for this study are protected due to subject privacy.

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
