# Peer review of "The Role of Distance from Home to Hospital on Parental Experience in the NICU: A Qualitative Study"

_children, 2023, doi:10.3390/children10091576_

Round 1
Reviewer 1 Report
I had a hard time navigating through the study. The study design can be improved for the questions that were intended to be studied ( to explore facilitators and barriers). The authors do provide a " Semi-Structured Interview guide", I am not sure if these are all the questions asked, were all the participants asked all these questions. It is also not exactly how the information was extracted from participants during the interview process since there is not a fixed questionnaire it seems. The authors have not mentioned how the participants were compensated. The authors mentioned various facilitators and barriers identified based on Quotes from participants, which may not be the ideal way of data collection.
Presented as such, this data is very hard to interpret.
acceptable
Author Response
We appreciate Reviewer 1’s feedback and thoughtful comments about how to improve the presentation of our study. As this was a qualitative study, we adhered to qualitative research methods, specifically adapted grounded theory which involves entering into interviews with participants with an established framework for parental engagement in the NICU setting. Individual in-depth interviews are widely used approaches for collecting qualitative data, particularly when seeking to foster learning about individual experiences and perspectives on a given set of issues. Our study focused on learning about individuals’ perspectives on facilitators and barriers to parental engagement in the NICU setting. In these 1:1 qualitative interviews, we used a semi-structured interview guide to ensure that the same topics were covered with each participant, while still allowing for individual and spontaneous dialogue between the interviewer and participant. This approach is well accepted in qualitative research and has been described by Kallio, et al. (PMID 27221824). These authors note that the “rigorous development of a qualitative semi-structured interview guide contributes to the objectivity and trustworthiness of studies and makes the results more plausible”. Peters and Halcomb (PMID 25783145) note that “semi-structured interviews, where the researcher has some predefined questions or topics but then probes further as the participant responds, can produce powerful data that provides insights into the participants’ experiences, perceptions or opinions.” We have included these references in our methods to help readers, who may be less familiar with qualitative research methods, better understand the framework and research process used for this study.
The methods, “Participants, Context and Recruitment” section was edited to clarify how participants were compensated, which was with a $40USD gift card. This information has been included on page 2, line 80.
Reviewer 2 Report
The presented manuscript is on a topic that I do believe is not so widely discussed but very important taking in mind the increasing percentage of pretertm infants born in recent years.
I do have some recommendations.
-The abstract can be improved because it focuses mainly on the methodology but it has almost no results so I suggest the authors to revise it and make it more concise and informative.
- In the manuscript on row 35 there is a citation which is not presented in the righ way. Please, do revise it.
-The present manuscript presents more specifically the situation of parents of VPT born in the USA and the specific problems that parents in the USA are facing with this matter but in the discussion I would have liked to see more referencing of international works if available to make the work more interesting to the readers and to be comparable as much as possible.
-Overall I thank to the authors for the interesting topic and the manuscript presented.
Author Response
The presented manuscript is on a topic that I do believe is not so widely discussed but very important taking in mind the increasing percentage of preterm infants born in recent years.
I do have some recommendations.
-The abstract can be improved because it focuses mainly on the methodology but it has almost no results so I suggest the authors to revise it and make it more concise and informative.
We appreciate Reviewer 2’s suggestion and have highlighted the facilitators and barriers as the results in the abstract. Due to the limited character count in the brief unstructured abstract format, it is challenging to expand upon all the individual themes.
- In the manuscript on row 35 there is a citation which is not presented in the righ way. Please, do revise it.
Thank you for noticing this error. Citation number 6 on line 25, page 1, has been corrected into the proper format.
-The present manuscript presents more specifically the situation of parents of VPT born in the USA and the specific problems that parents in the USA are facing with this matter but in the discussion I would have liked to see more referencing of international works if available to make the work more interesting to the readers and to be comparable as much as possible.
Thank you for this suggestion. We have amended the discussion to include international literature on the use of technology in the NICU, and to highlight the differential impact based on country of residence for families who reside in countries with different family leave policies.
-Overall I thank to the authors for the interesting topic and the manuscript presented.
Reviewer 3 Report
The objective of this study was to explore the impact of distance from home to hospital on the engage of parents of VPT infants in the NICU.
Manuscript is clear, relevant for the field, scientifically sound and presented in a well-structured manner. The manuscript’s results are reproducible based on the details given in the methods section. The conclusions are consistent with the evidence and arguments presented.
Author Response
The objective of this study was to explore the impact of distance from home to hospital on the engage of parents of VPT infants in the NICU.
Manuscript is clear, relevant for the field, scientifically sound and presented in a well-structured manner. The manuscript’s results are reproducible based on the details given in the methods section. The conclusions are consistent with the evidence and arguments presented.
Thank you, Reviewer 3, for the positive feedback on our manuscript.